# Single Path One-Shot Neural Architecture Search with Uniform Sampling

## Abstract

We revisit the one-shot Neural Architecture Search (NAS) paradigm and analyze its advantages over existing NAS approaches. Existing one-shot method (Bender et al., 2018), however, is hard to train and not yet effective on large scale datasets like ImageNet. This work propose a Single Path One-Shot model to address the challenge in the training. Our central idea is to construct a simplified supernet, where all architectures are single paths so that weight co-adaption problem is alleviated. Training is performed by uniform path sampling. All architectures (and their weights) are trained fully and equally.

Comprehensive experiments verify that our approach is flexible and effective. It is easy to train and fast to search. It effortlessly supports complex search spaces (e.g., building blocks, channel, mixed-precision quantization) and different search constraints (e.g., FLOPs, latency). It is thus convenient to use for various needs. It achieves start-of-the-art performance on the large dataset ImageNet.

## 1 Introduction

Deep learning automates *feature engineering* and solves the *weight optimization* problem. Neural Architecture Search (NAS) aims to automate *architecture engineering* by solving one more problem, *architecture design*. Early NAS approaches (Zoph et al., 2018; Zhong et al., 2018a;b; Liu et al., 2018a; Real et al., 2018; Tan et al., 2018) solves the two problems in a *nested* manner. A large number of architectures are sampled and trained from scratch. The computation cost is unaffordable on large datasets.

Recent approaches (Wu et al., 2018a; Cai et al., 2018; Liu et al., 2018b; Xie et al., 2018; Pham et al., 2018; Zhang et al., 2018c; Brock et al., 2017; Bender et al., 2018) adopt a *weight sharing* strategy to reduce the computation. A supernet subsuming all architectures is trained only once. Each architecture inherits its weights from the supernet. Only fine-tuning is performed. The computation cost is greatly reduced.

Most weight sharing approaches use a continuous relaxation to parameterize the search space (Wu et al., 2018a; Cai et al., 2018; Liu et al., 2018b; Xie et al., 2018; Zhang et al., 2018c). The architecture distribution parameters are *jointly* optimized during the supernet training via gradient based methods. The best architecture is sampled from the distribution after optimization. There are two issues in this formulation. *First*, the weights in the supernet are deeply coupled. It is unclear why inherited weights for a specific architecture are still effective. *Second*, joint optimization introduces further coupling between the architecture parameters and supernet weights. The greedy nature of the gradient based methods inevitably introduces bias during optimization and could easily mislead the architecture search. They adopted complex optimization techniques to alleviate the problem.

The one-shot paradigm (Brock et al., 2017; Bender et al., 2018) alleviates the second issue. It defines the supernet and performs weight inheritance in a similar way. However, there is no architecture relaxation. The architecture search problem is decoupled from the supernet training and addressed in a separate step. Thus, it is *sequential*. It combines the merits of both *nested* and *joint* optimization approaches above. The architecture search is both efficient and flexible.

The first issue is still problematic. Existing one-shot approaches (Brock et al., 2017; Bender et al., 2018) still have coupled weights in the supernet. Their optimization is complicated and involves sensitive hyper parameters. They have not shown competitive results on large datasets.

This work revisits the one-shot paradigm and presents a new approach that further eases the training and enhances architecture search. Based on the observation that the accuracy of an architecture using inherited weights should be predictive for the accuracy using optimized weights, we propose that the supernet training should be *stochastic*. All architectures have their weights optimized simultaneously. This gives rise to a *uniform sampling* strategy. To reduce the weight coupling in the supernet, a simple search space that consists of *single path* architectures is proposed. The training is hyperparameter-free and easy to converge.

This work makes the following contributions.

1. We present a principled analysis and point out drawbacks in existing NAS approaches that use nested and joint optimization. Consequently, we hope this work will renew interest in the one-shot paradigm, which combines the merits of both via sequential optimization.

2. We present a single path one-shot approach with uniform sampling. It overcomes the drawbacks of existing one-shot approaches. Its simplicity enables a rich search space, including novel designs for channel size and bit width, all addressed in a unified manner. Architecture search is efficient and flexible. Evolutionary algorithm is used to support real world constraints easily, such as low latency.

Comprehensive ablation experiments and comparison to previous works on a large dataset (ImageNet) verify that the proposed approach is state-of-the-art in terms of accuracy, memory consumption, training time, architecture search efficiency and flexibility.

## 2    REVIEW OF NAS APPROACHES

Without loss of generality, the architecture search space $\mathcal{A}$ is represented by a directed acyclic graph (DAG). A network architecture is a subgraph $a \in \mathcal{A}$, denoted as $\mathcal{N}(a, w)$ with weights $w$.

Neural architecture search aims to solve two related problems. The first is *weight optimization*,

$$w_a = \arg\min_{w} \mathcal{L}_{\text{train}}\left(\mathcal{N}(a, w)\right),\tag{1}$$

where $\mathcal{L}_{\text{train}}(\cdot)$ is the loss function on the training set.

The second is *architecture optimization*. It finds the architecture that is trained on the training set and has the best accuracy on the validation set, as

$$a^* = \arg\max_{a \in \mathcal{A}} \text{ACC}_{\text{val}}\left(\mathcal{N}(a, w_a)\right),\tag{2}$$

where $\text{ACC}_{\text{val}}(\cdot)$ is the accuracy on the validation set.

Early NAS approaches perform the two optimization problems in a *nested* manner (Zoph & Le, 2016; Zoph et al., 2018; Zhong et al., 2018a;b; Baker et al., 2016). Numerous architectures are sampled from $\mathcal{A}$ and trained from scratch as in Eq. (1). Each training is expensive. Only small dataset (e.g., CIFAR 10) and small search space (e.g, a single block) are affordable.

Recent NAS approaches adopt a *weight sharing* strategy (Cai et al., 2018; Liu et al., 2018b; Wu et al., 2018a; Xie et al., 2018; Bender et al., 2018; Brock et al., 2017; Zhang et al., 2018c; Pham et al., 2018). The architecture search space $\mathcal{A}$ is encoded in a *supernet*[1], denoted as $\mathcal{N}(\mathcal{A}, W)$, where $W$ is the weights in the supernet. The supernet is trained once. All architectures inherit their weights directly from $W$. Thus, they share the weights in their common graph nodes. Fine tuning of an architecture is performed in need, but no training from scratch is incurred. Therefore, architecture search is fast and suitable for large datasets like ImageNet.

Most weight sharing approaches convert the discrete architecture search space into a continuous one (Wu et al., 2018a; Cai et al., 2018; Liu et al., 2018b; Xie et al., 2018; Zhang et al., 2018c). Formally, space $\mathcal{A}$ is relaxed to $\mathcal{A}(\theta)$, where $\theta$ denotes the continuous parameters that represent the *distribution* of the architectures in the space. Note that the new space subsumes the original one, $\mathcal{A} \subseteq \mathcal{A}(\theta)$. An architecture sampled from $\mathcal{A}(\theta)$ could be invalid in $\mathcal{A}$.

---

[1]"Supernet" is used as a general concept in this work. It has different names and implementation in previous approaches.

An advantage of the continuous search space is that gradient based methods (Liu et al., 2018b; Cai et al., 2018; Wu et al., 2018a; Véniat & Denoyer, 2018; Xie et al., 2018; Zhang et al., 2018c) is feasible. Both weights and architecture distribution parameters are *jointly* optimized, as

$$(\theta^*, W_{\theta^*}) = \arg\min_{\theta, W} \mathcal{L}_{train}(\mathcal{N}(\mathcal{A}(\theta), W)). \tag{3}$$

After optimization, the best architecture $a^*$ is sampled from $\mathcal{A}(\theta^*)$. Note that it could be invalid in $\mathcal{A}$. If so, it is validated (e.g., by binarization of $\theta$ (Liu et al., 2018b)). It then inherits the weights from $W_{\theta^*}$ and is fine-tuned.

Optimization of Eq. (3) is challenging. *First*, the weights of the graph nodes in the supernet depend on each other and become *deeply coupled* during optimization. For a specific architecture, it inherits certain node weights from $W$. While these weights are decoupled from the others, it is unclear why they are still effective.

*Second*, joint optimization of architecture parameter $\theta$ and weights $W$ introduces further complexity. Solving Eq. (3) inevitably introduces bias to certain areas in $\theta$ and certain nodes in $W$ during the progress of optimization. The bias would leave some nodes in the graph well trained and others poorly trained. With different level of maturity in the weights, different architectures are actually non-comparable. However, their prediction accuracy is used as guidance for sampling in $\mathcal{A}(\theta)$ (e.g., used as reward in policy gradient (Cai et al., 2018)). This would further mislead the architecture sampling. This problem is in analogy to the "dilemma of exploitation and exploration" problem in reinforcement learning. To alleviate such problems, existing approaches adopt complicated optimization techniques (see Table 7 for a summary). Nevertheless, there lacks a comprehensive evaluation of their effectiveness (Li & Talwalkar, 2019).

**Task constraints**   Real world tasks usually have additional requirements on a network's memory consumption, FLOPs, latency, energy consumption, etc. These requirements only depends on the architecture $a$, not on the weights $w_a$. Thus, they are called *architecture constraints* in this work. A typical constraint is that the network's latency is no more than a preset budget, as

$$\text{Latency}(a^*) \leq \text{Lat}_{\max}. \tag{4}$$

Note that it is challenging to satisfy Eq. (2) and Eq. (4) simultaneously for most previous approaches. Some works augment the loss function $\mathcal{L}_{train}$ in Eq. (3) with *soft* loss terms that consider the architecture latency (Cai et al., 2018; Wu et al., 2018a; Xie et al., 2018; Véniat & Denoyer, 2018). However, it is hard, if not impossible, to guarantee a hard constraint like Eq. (4).

## 3   OUR SINGLE PATH ONE-SHOT APPROACH

As analyzed above, the coupling between architecture parameters and weights is problematic. This is caused by joint optimization of both. To alleviate the problem, a natural solution is to *decouple* the super net training and architecture search in two *sequential* steps. This leads to the so called *one-shot* approaches (Brock et al., 2017; Bender et al., 2018).

In general, the two steps are formulated as follows. Firstly, the supernet weight is optimized as

$$W_{\mathcal{A}} = \arg\min_{W} \mathcal{L}_{\text{train}}\left(\mathcal{N}(\mathcal{A}, W)\right). \tag{5}$$

Compared to Eq. (3), the continuous parameterization of search space is absent. Only weights are optimized.

Secondly, architecture searched is performed as

$$a^* = \arg\max_{a \in \mathcal{A}} \text{ACC}_{\text{val}}\left(\mathcal{N}(a, W_{\mathcal{A}}(a))\right). \tag{6}$$

During search, each sampled architecture $a$ inherits its weights from $W_{\mathcal{A}}$ as $W_{\mathcal{A}}(a)$. The key difference of Eq. (6) from Eq. (1) and (2) is that architecture weights are ready for use. Evaluation of $\text{ACC}_{val}(\cdot)$ only requires inference. Thus, the search is very *efficient*.

The search is also *flexible*. Any adequate search algorithm is feasible. The architecture constraint like Eq. (4) can be exactly satisfied. Search can be repeated many times on the same supernet once

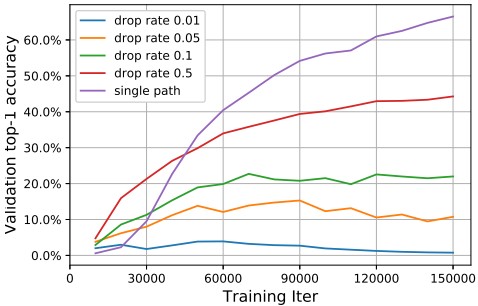 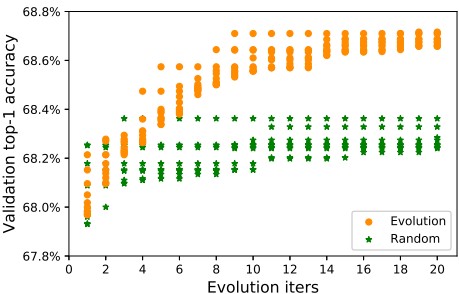

Figure 1: Comparison of single path strategy and drop path strategy

Figure 2: Evolutionary vs. random architecture search.

trained, using different constraints (e.g., 100ms latency and 200ms latency). These properties are absent in previous approaches. These make the one-shot paradigm attractive for real world tasks.

One problem in Sec. 2 still remains. The graph nodes' weights in the supernet training in Eq.( 5) are coupled. It is unclear why the inherited weights $W_{\mathcal{A}}(a)$ are still good for an arbitrary architecture $a$.

The recent one-shot approach (Bender et al., 2018) attempts to decouple the weights using a "path dropout" strategy. During an SGD step in Eq. (5), each edge in the supernet graph is randomly dropped. The random chance is controlled via a dropout rate parameter. In this way, the co-adaptation of the node weights is reduced during training. Experiments in (Bender et al., 2018) indicate that the training is very sensitive to the dropout rate parameter. This makes the supernet training hard. A carefully tuned heat-up strategy is used. In our implementation of this work, we also found that the validation accuracy is very sensitive to the dropout rate parameter.

**Single Path Supernet and Uniform Sampling.** Let us restart to think about the fundamental principle behind the idea of weight sharing. The key to the success of architecture search in Eq. (6) is that, the accuracy of $any$ architecture $a$ on a validation set using inherited weight $W_{\mathcal{A}}(a)$ (without extra fine tuning) is highly predictive for the accuracy of $a$ that is fully trained. Ideally, this requires that the weight $W_{\mathcal{A}}(a)$ to approximate the optimal weight $w_a$ as in Eq. (1). The quality of the approximation depends on how well the training loss $\mathcal{L}_{\text{train}}(\mathcal{N}(a, W_{\mathcal{A}}(a)))$ is minimized. This gives rise to the principle that *the supernet weights $W_{\mathcal{A}}$ should be optimized in a way that all architectures in the search space are optimized simultaneously*. This is expressed as

$$W_{\mathcal{A}} = \underset{W}{\arg\min}\, \mathbb{E}_{a \sim \Gamma(\mathcal{A})} \left[ \mathcal{L}_{\text{train}}(\mathcal{N}(a, W(a))) \right], \tag{7}$$

where $\Gamma(\mathcal{A})$ is a prior distribution of $a \in \mathcal{A}$. Note that Eq. (7) is an implementation of Eq. (5). In each step of optimization, an architecture $a$ is randomly sampled. Only weights $W(a)$ are activated and updated. So the memory usage is efficient. In this sense, the supernet is no longer a valid network. It behaves as a *stochastic supernet* (Véniat & Denoyer, 2018). This is different from (Bender et al., 2018).

To reduce the co-adaptation between node weights, we propose a supernet structure that each architecture is a *single path*, as shown in Fig.3. Compared to the path dropout strategy in (Bender et al., 2018), the single path strategy is hyperparameter-free. We compared the two strategies within the same search space (as in this work). Note that the original *drop path* in (Bender et al., 2018) may drop all operations in a block, resulting in a short cut of identity connection. In our implementation, it is forced that one random path is kept in this case since our choice block does not have an identity branch. We randomly select sub network and evaluate its validation accuracy during the training stage. Results in Fig.1 show that drop rate parameters matters a lot. Our single path strategy corresponds to using drop rate 1. It works the best, which also verifies the benefits of weight decoupling by our single path strategy.

The prior distribution $\Gamma(\mathcal{A})$ is important. In this work, we empirically find that *uniform sampling* is good. This is not much of a surprise. A recent work also finds that purely random search is competitive to several SOTA NAS approaches( Li & Talwalkar (2019)). We also experimented with

a variant that samples the architectures uniformly according to their constraints, named uniform constraint sampling. Specifically, we randomly choose a range, and then sample the architecture repeatedly until the FLOPs of sampled architecture falls in the range. This is because a real task usually expects to find multiple architectures satisfying different constraints. In this work, we find the uniform constraint sampling method is slightly better. So we use it by default in this paper.

We note that sampling a path according to architecture distribution during optimization is already used in previous weight sharing approaches (Pham et al., 2018; Véniat & Denoyer, 2018; Wu et al., 2018a; Cai et al., 2018; Xie et al., 2018; Zhang et al., 2018c; Yao et al., 2019; Dong & Yang, 2019; Stamoulis et al., 2019). The difference is that, the distribution $\Gamma(\mathcal{A})$ is a *fixed* priori during our training (Eq. (7)), while it is *learnable and updated* (Eq. (3)) in previous approaches (e.g. RL (Pham et al., 2018), policy gradient (Véniat & Denoyer, 2018; Cai et al., 2018), Gumbel Softmax (Wu et al., 2018a; Xie et al., 2018), APG (Zhang et al., 2018c)). As analyzed in Sec. 2, the latter makes the supernet weights and architecture parameters highly correlated and optimization difficult.

Comprehensive experiments in Sec. 4 show that our approach achieves better results than the SOTA methods. Note that there is no such theoretical guarantee that using a fixed prior distribution is *inherently* better than optimizing the distribution during training. Our better result likely indicates that the joint optimization in Eq. (3) is too difficult for the existing optimization techniques.

**Supernet Architecture and Novel Choice Block Design.** *Choice blocks* are used to build a *stochastic* architecture. Fig. 3 illustrates an example case. A choice block consists of multiple architecture choices. For our single path supernet, each choice block only has one choice invoked at the same time. A path is obtained by sampling all the choice blocks.

The simplicity of our approach enables us to define different types of choice blocks to search various architecture variables. Specifically, we propose two novel choice blocks to support complex search spaces.

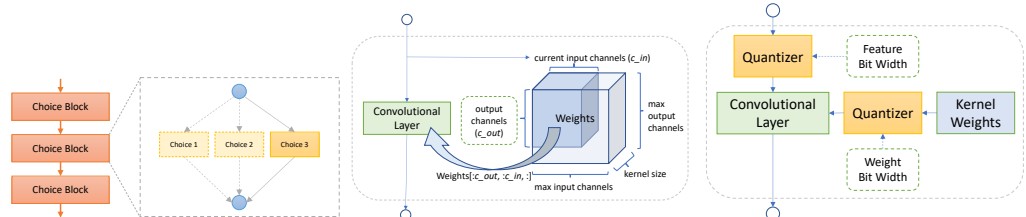

Figure 3: *Choice blocks* for our single path supernet.

Figure 4: *Choice block* for channel number search.

Figure 5: *Choice block* for mixed-precision quantization search.

*Channel Number Search.* We propose a new choice block based on weight sharing, as shown in Fig. 4. The main idea is to preallocate a weight tensor with maximum number of channels, and the system randomly selects the channel number and slices out the corresponding subtensor for convolution. With the weight sharing strategy, we found that the supernet can converge quickly.

In detail, assume the dimensions of preallocated weights are (max_c_out, max_c_in, ksize). For each batch in supernet training, the number of current output channels $c\_out$ is randomly sampled. Then, we slice out the weights for current batch with the form $\text{Weights}[: c\_out, : c\_in, :]$, which is used to produce the output. The optimal number of channels is determined in the search step.

*Mixed-Precision Quantization Search.* In this work, We design a novel *choice block* to search the bit widths of the weights and feature maps, as shown in Fig. 5. We also combine the *channel search space* discussed earlier to our *mixed-precision quantization search space*. During supernet training, for each choice block feature bit width and weight bit width are randomly sampled. They are determined in the evolutionary step. See Sec. 4 and Fig. 5 for details.

**Evolutionary Architecture Search.** For architecture search in Eq. (6), previous one-shot works (Brock et al., 2017; Bender et al., 2018) use random search. This is not effective for a large search space. This work uses an evolutionary algorithm. Note that evolutionary search in NAS is used in (Real et al., 2018), but it is costly as each architecture is trained from scratch. In our search, each architecture only performs inference. This is very efficient.

The algorithm is elaborated in Algorithm 1. For all experiments, population size $P = 50$, max iterations $\mathcal{T} = 20$ and $k = 10$. For crossover, two randomly selected candidates are crossed to produce a new one. For mutation, a randomly selected candidate mutates its every choice block with probability 0.1 to produce a new candidate. Crossover and mutation are repeated to generate enough new candidates that meet the given architecture constraints. Before the inference of an architecture, the statistics of all the *Batch Normalization (BN)* (Ioffe & Szegedy, 2015) operations are recalculated on a random subset of training data (20000 images on ImageNet). It takes a few seconds. This is because the BN statistics from the supernet are usually not applicable to the candidate nets. This is also referred in (Bender et al., 2018).

Fig. 2 plots the validation accuracy over generations, using both evolutionary and random search methods. It is clear that evolutionary search is more effective. Experiment details are in Sec. 4.

The evolutionary algorithm is flexible in dealing with different constraints in Eq. (4), because the mutation and crossover processes can be directly controlled to generate proper candidates to satisfy the constraints. Previous RL-based (Tan et al., 2018) and gradient-based (Cai et al., 2018; Wu et al., 2018a; Véniat & Denoyer, 2018) methods design tricky rewards or loss functions to deal with such constraints. For example, (Wu et al., 2018a) uses a loss function $\mathrm{CE}(a, w_a) \cdot \alpha \log(\mathrm{LAT}(a))^\beta$ to balance the accuracy and the latency. It is hard to tune the hyper parameter $\beta$ to satisfy a hard constraint like Eq. (4).

**Summary.** The combination of single path supernet, uniform sampling training strategy, evolutionary architecture search, and rich search space design makes our approach simple, efficient and flexible. Table 7 in Appendix performs a comprehensive comparison of our approach against previous weight sharing approaches on various aspects. Ours is the easiest to train, occupies the smallest memory, best satisfies the architecture (latency) constraint, and easily supports large datasets. Extensive results in Sec. 4 verify that our approach is the state-of-the-art.

## 4 EXPERIMENT RESULTS

**Dataset.** All experiments are performed on *ImageNet* (Russakovsky et al., 2015). We randomly split the original training set into two parts: 50000 images for validation (50 images for each class exactly) and the rest as the training set. The original validation set is used for testing, on which all the evaluation results are reported, following (Cai et al., 2018).

**Training.** For the training of the supernet and retraining of the best architecture (after evolutionary search) from scratch, we use the same settings (including data augmentation strategy, learning rate schedule, etc.) as (Ma et al., 2018). The batch size is 1024. Supernet is trained for 120 epochs (150000 iterations) and the best architecture for 240 epochs (300000 iterations). Training uses 8 *NVIDIA GTX 1080Ti* GPUs.

**Search Space: Building Blocks.** First, we evaluate our method on the task of *building block selection*, i.e. to find the optimal combination of building blocks under a certain complexity constraint. Our basic building block design is inspired by a state-of-the-art manually-designed network – *ShuffleNet v2* (Ma et al., 2018). Table 1 shows the overall architecture of the supernet. There are 20 *choice blocks* in total. Each choice block has 4 candidates, namely "choice_3", "choice_5", "choice_7" and "choice_x" respectively (see Fig.6 in Appendix for details). They differ in kernel sizes and the number of depthwise convolutions. The size of the search space is $4^{20}$.

We use FLOPs $\leq 330M$ as the complexity constraint, as the FLOPs of a plenty of previous networks lies in [300,330], including manually-designed networks (Howard et al., 2017; Sandler et al., 2018; Zhang et al., 2018b; Ma et al., 2018) and those obtained in NAS (Cai et al., 2018; Wu et al., 2018a; Tan et al., 2018).

Table 2 shows the results. For comparison, we set up a series of baselines as follows: 1) select a certain block choice only (denoted by "all choice_*" entries); note that different choices have different FLOPs, thus we adjust the channels to meet the constraint. 2) Randomly select some candidates from the search space. 3) Replace our evolutionary architecture optimization with random search used in (Brock et al., 2017; Bender et al., 2018). Results show that random search equipped with our single path supernet finds an architecture only slightly better that random select (73.8 vs. 73.7). It does no mean that our single path supernet is less effective. This is because the random search is too

| input shape | block | channels | repeat | stride |
|---|---|---|---|---|
| $224^2 \times 3$ | $3 \times 3$ conv | 16 | 1 | 2 |
| $112^2 \times 16$ | CB | 64 | 4 | 2 |
| $56^2 \times 64$ | CB | 160 | 4 | 2 |
| $28^2 \times 160$ | CB | 320 | 8 | 2 |
| $14^2 \times 320$ | CB | 640 | 4 | 2 |
| $7^2 \times 640$ | $1 \times 1$ conv | 1024 | 1 | 1 |
| $7^2 \times 1024$ | GAP | - | 1 | - |
| 1024 | fc | 1000 | 1 | - |

| model | FLOPs | top-1 acc(%) |
|---|---|---|
| all choice_3 | 324M | 73.4 |
| all choice_5 | 321M | 73.5 |
| all choice_7 | 327M | 73.6 |
| all choice_x | 326M | 73.5 |
| random select (5 times) | ∼320M | ∼73.7 |
| SPS + random search | 323M | 73.8 |
| ours (fully-equipped) | 319M | **74.3** |

Table 1: Supernet architecture. *CB* - choice block. *GAP* - global average pooling. "stride" column represents the stride of the first block in each repeated group.

Table 2: Results of building block search. *SPS* The – single path supernet.

naive to pick good candidates from the large search space. Using evolutionary search, our approach finds out an architecture that achieves superior accuracy (74.3) over all the baselines.

**Search Space: Channels.** Based on our novel choice block for channel number search, we first evaluate channel search on the baseline structure "all choice_3" (refer to Table 2): for each building block, we search the number of "mid-channels" (output channels of the first 1x1 conv in each building block) varying from 0.2x to 1.6x (with stride 0.2), where "$k$-x" means $k$ times the number of default channels. Same as Sec. 4, we set the complexity constraint FLOPs $\leq 330M$. Table 3 (first part) shows the result. Our channel search method has higher accuracy (73.9) than the baselines.

To further boost the accuracy, we search building blocks and channels jointly. There are two alternatives: 1) running channel search on the best building block search result of Sec. 4; or 2) searching on the combined search space directly. In our experiments, we find the results of the first pipeline is slightly better. As shown in Table 3, searching in the joint space achieves the best accuracy (**74.7%** acc.), surpassing all the previous state-of-the-art manually-designed (Ma et al., 2018; Sandler et al., 2018) or automatically-searched models (Tan et al., 2018; Zoph et al., 2018; Liu et al., 2018a;b; Cai et al., 2018; Wu et al., 2018a) under the complexity of ∼ 300M FLOPs.

**Comparison with State-of-the-arts.** Results in Table 3 shows our method is superior. Nevertheless, the comparisons could be unfair because different search spaces and training methods are used in previous works (Cai et al., 2018). To make *direct* comparisons, we benchmark our approach to the *same* search space of (Cai et al., 2018; Wu et al., 2018a). In addition, we retrain the searched models reported in (Cai et al., 2018; Wu et al., 2018a) under the same settings to guarantee the fair comparison.

The search space and supernet architecture in *ProxylessNAS* (Cai et al., 2018) is inspired by *MobileNet v2* (Sandler et al., 2018) and *MnasNet* (Tan et al., 2018). It contains 21 *choice blocks*; each choice block has 7 choices (6 different building blocks and one skip layer). The size of the search space is $7^{21}$. *FBNet* (Wu et al., 2018a) also uses a similar search space.

| Model | FLOPs/Params | Top-1 acc(%) |
|---|---|---|
| all choice_3 | 324M/3.1M | 73.4 |
| rand sel. channels (5 times) | ∼ 323M/3.2M | ∼ 73.1 |
| choice_3 + channel search | 329M/3.4M | **73.9** |
| rand sel. blocks + channels | ∼ 325M/3.2M | ∼ 73.4 |
| block search | 319M/3.3M | 74.3 |
| block search + channel search | 328M/3.4M | **74.7** |
| MobileNet V1 (0.75x) Howard et al. (2017) | 325M/2.6M | 68.4 |
| MobileNet V2 (1.0x) Sandler et al. (2018) | 300M/3.4M | 72.0 |
| ShuffleNet V2 (1.5x) Ma et al. (2018) | 299M/3.5M | 72.6 |
| NASNET-A Zoph et al. (2018) | 564M/5.3M | 74.0 |
| PNASNET Liu et al. (2018a) | 588M/5.1M | 74.2 |
| MnasNet Tan et al. (2018) | 317M/4.2M | 74.0 |
| DARTS Liu et al. (2018b) | 595M/4.7M | 73.1 |
| Proxyless-R (mobile)* Cai et al. (2018) | 320M/4.0M | 74.2 (74.6) |
| FBNet-B* Wu et al. (2018a) | 295M/4.5M | 74.1 (74.1) |

Table 3: Results of channel search. * Performances are reported in the form "x (y)", where "x" means the accuracy retrained by us and "y" means accuracy reported by the original paper.

Table 4 reports the accuracy and complexities (FLOPs and latency on our device) of 5 models searched by (Cai et al., 2018; Wu et al., 2018a), as the baselines. Then, for each baseline, our search method runs under the constraints of same FLOPs or same latency, respectively. Results shows that for all the cases our method achieves comparable or higher accuracy than the counterpart baselines. We also point out that since the target devices in (Cai et al., 2018; Wu et al., 2018a) are different from ours, the reported results may be sub-optimal on our platform.

Furthermore, it is worth noting that all our 10 architectures in Table 4 are searched on the *same* supernet, justifying the flexibility and efficiency of our approach to deal with different complexity

constraints: supernet is trained once and searched multiple times. In contrast, previous methods (Wu et al., 2018a; Cai et al., 2018) have to train multiple supernets under various constraints. According to Table 6, searching is much cheaper than supernet training.

| baseline network | FLOPs | latency | top-1 acc(%) baseline | top-1 acc(%) ours (same FLOPs) | top-1 acc(%) ours (same latency) |
|---|---|---|---|---|---|
| FBNet-A (Wu et al., 2018a) | 249M/4.3M | 13ms | 73.0 (73.0) | **73.2** | **73.3** |
| FBNet-B (Wu et al., 2018a) | 295M/4.5M | 17ms | 74.1 (74.1) | **74.2** | **74.8** |
| FBNet-C (Wu et al., 2018a) | 375M/5.5M | 19ms | 74.9 (74.9) | **75.0** | **75.1** |
| Proxyless-R (mobile) (Cai et al., 2018) | 320M/4.0M | 17ms | 74.2 (74.6) | **74.5** | **74.8** |
| Proxyless (GPU) (Cai et al., 2018) | 465M/5.3M | 22ms | 74.7 (75.1) | **74.8** | **75.3** |

Table 4: Compared with state-of-the-art *NAS* methods (Wu et al., 2018a; Cai et al., 2018) *using the same search space*. The latency is evaluated on a single *NVIDIA Titan XP* GPU, with $batchsize = 32$. Accuracy numbers in the brackets are reported by the original papers; others are trained by us. All our architectures are searched from the **same** supernet via evolutionary architecture optimization.

**Application: Mixed-Precision Quantization.**
We evaluate our method on *ResNet-18* and *ResNet-34* as common practice in previous quantization works (e.g. (Choi et al., 2018; Wu et al., 2018b; Liu et al., 2018c; Zhou et al., 2016; Zhang et al., 2018a)). Following (Zhou et al., 2016; Choi et al., 2018; Wu et al., 2018b), we only search and quantize the *res-blocks*, excluding the first convolutional layer and the last fully-connected layer. In the search space, choices of weight and feature bit widths include $\{(1, 2), (2, 2), (1, 4), (2, 4), (3, 4), (4, 4)\}$. As for channel search, we search the number of "bottleneck channels" (i.e. the output channels of the first convolutional layer in each residual

| Method | BitOPs | top1-acc(%) | Method | BitoPs | top1-acc(%) |
|---|---|---|---|---|---|
| ResNet-18 | float point | 70.9 | ResNet-34 | float point | 75.0 |
| 2W2A | 6.32G | 65.6 | 2W2A | 13.21G | 70.8 |
| ours | **6.21G** | **66.4** | ours | **13.11G** | **71.5** |
| 3W3A | 14.21G | 68.3 | 3W3A | 29.72G | 72.5 |
| DNAS | 15.62G | 68.7 | DNAS | 38.64G | 73.2 |
| ours | **13.49G** | **69.4** | ours | **28.78G** | **73.9** |
| 4W4A | 25.27G | 69.3 | 4W4A | 52.83G | 73.5 |
| DNAS | 25.70G | **70.6** | DNAS | 57.31G | 74.0 |
| ours | **24.31G** | 70.5 | ours | **51.92G** | **74.6** |

Table 5: Results of mixed-precision quantization search. "$k$W$k$A" means $k$-bit quantization for all the weights and activations. DNAS (Wu et al., 2018b).

block) in $\{0.5x, 1.0x, 1.5x\}$, where "$k$-x" means $k$ times the number of original channels. The size of the search space is $(3 \times 6)^N = 18^N$, where $N$ is the number of choice blocks ($N = 8$ for ResNet-18 and $N = 16$ for ResNet-34). Note that for each building block we use the same bit widths for the two convolutions. We use *PACT* (Choi et al., 2018) as the quantization algorithm.

Table 5 reports the results. The baselines are denoted as $k$W$k$A ($k = 2, 3, 4$), which means uniform quantization of weights and activations with $k$-bits. Then, our search method runs under the constraints of the corresponding BitOps. We also compare with a recent mixed-precision quantization search approach (Wu et al., 2018b). Results shows that our method achieves superior accuracy in most cases. Also note that all our results for ResNet-18 and ResNet-34 are searched on the **same** supernet. This is very efficient.

**Search Cost Analysis.** The search cost is a matter of concern in NAS methods. So we analyze the search cost of our method and previous methods (Wu et al., 2018a; Cai et al., 2018) (reimplemented by us). We use the search space of our *building blocks* to measure the memory cost of training supernet and overall time cost. All the supernets are trained for 150000 iterations with a batch size of 256. All models are trained with 8 GPUs. The Table 6 shows that our approach clearly uses less memory than

| Method | Proxyless | FBNet | Ours |
|---|---|---|---|
| GPU memory cost (8 GPUs in total) | 37G | 63G | 24G |
| Training time | 15 Gds | 20 Gds | 12 Gds |
| Search time | 0 | 0 | <1 Gds |
| Retrain time | 16 Gds | 16 Gds | 16 Gds |
| Total time | 31 Gds | 36 Gds | 29 Gds |

Table 6: Search Cost. *Gds* - GPU days

other two methods because of the single path supernet. And our approach is much more efficient overall although we have an extra search step that costs less than 1 GPU day. Note Table 6 only compares a single run. In practice, our approach is more advantageous and more convenient to use when multiple searches are needed. As summarized in Table 7, it guarantees to find out the architecture satisfying constraints within one search. Repeated search is easily supported.

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

## A  APPENDIX

---

**Algorithm 1:** Evolutionary Architecture Search

---

1 **Input**: *supernet weights* $W_{\mathcal{A}}$, *population size P, architecture constraints* $\mathcal{C}$, *max iteration* $\mathcal{T}$, *validation dataset* $D_{val}$

2 **Output**: *the architecture with highest validation accuracy under architecture constraints*

3 $P_0 := Initialize\_population(P, \mathcal{C})$; $Topk := \emptyset$;

4 $n := P/2$;                                                                                          Crossover number

5 $m := P/2$;                                                                                          Mutation number

6 $prob := 0.1$;                                                                                       Mutation probability

7 **for** $i = 1 : \mathcal{T}$ **do**

8 |    $ACC_{i-1} := Inference(W_{\mathcal{A}}, D_{val}, P_{i-1})$;

9 |    $Topk := Update\_Topk(Topk, P_{i-1}, ACC_{i-1})$;

10 |   $P_{crossover} := Crossover(Topk, n, \mathcal{C})$;

11 |   $P_{mutation} := Mutation(Topk, m, prob, \mathcal{C})$;

12 |   $P_i := P_{crossover} \cup P_{mutation}$;

13 **end**

14 Return the architecture with highest accuracy in $Topk$;

---

| Approach | Supernet optimization | Architecture search | Hyper parameters in supernet Training | Memory consumption in supernet training | How to satisfy constraint | Experiment on ImageNet |
|---|---|---|---|---|---|---|
| ENAS (Pham et al., 2018) | Alternative RL and fine tuning | | Short-time fine tuning setting | Single path + RL system | None | No |
| BSN (Véniat & Denoyer, 2018) | Stochastic super networks + policy gradient | | Weight of cost penalty | Single path | Soft constraint in training. Not guaranteed | No |
| DARTS (Liu et al., 2018b) | Gradient-based, path dropout | | Path dropout rate. Weight of auxiliary loss | Whole supernet | None | Transfer |
| Proxyless (Cai et al., 2018) | Stochastic relaxation of the discrete search + policy gradient | | Scaling factor of latency loss | Two paths | Soft constraint in training. Not guaranteed. | Yes |
| FBNet (Wu et al., 2018a) | Stochastic relaxation of the discrete search to differentiable optimization via Gumbel softmax | | Temperature parameter in Gumbel softmax. Coefficient in constraint loss | Whole supernet | Soft constraint in training. Not guaranteed. | Yes |
| SNAS (Xie et al., 2018) | Same as FBNet | | Same as FBNet | Whole supernet | Soft constraint in training. Not guaranteed. | Transfer |
| SMASH (Brock et al., 2017) | Hypernetwork | Random | None | Hypernet+single Path | None | No |
| One-Shot (Bender et al., 2018) | Path dropout | Random | Drop rate | Whole supernet | Not investigated | Yes |
| Ours | Uniform path sampling | Evolution | None | Single path | Guaranteed in searching. Support multiple constraints. | Yes |

Table 7: Overview and comparison of SOTA *weight sharing* approaches. Ours is the easiest to train, occupies the smallest memory, best satisfy the architecture (latency) constraint, and easily supports the large dataset. Note that those approaches belonging to the joint optimization category (Eq. (3)) have "Supernet optimization" and "Architecture search" columns merged.

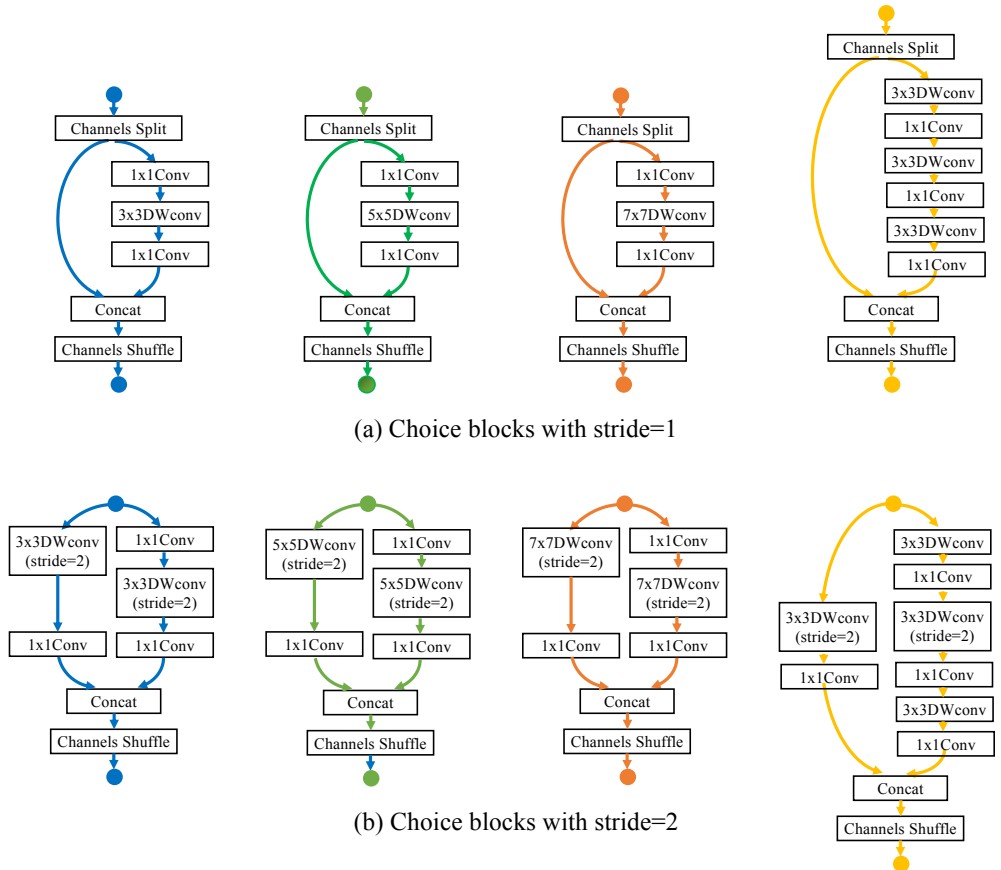

Figure 6: *Choice blocks* used in Sec. 4. From left to right : *Choice_3, Choice_5, Choice_7, Choice_x*.

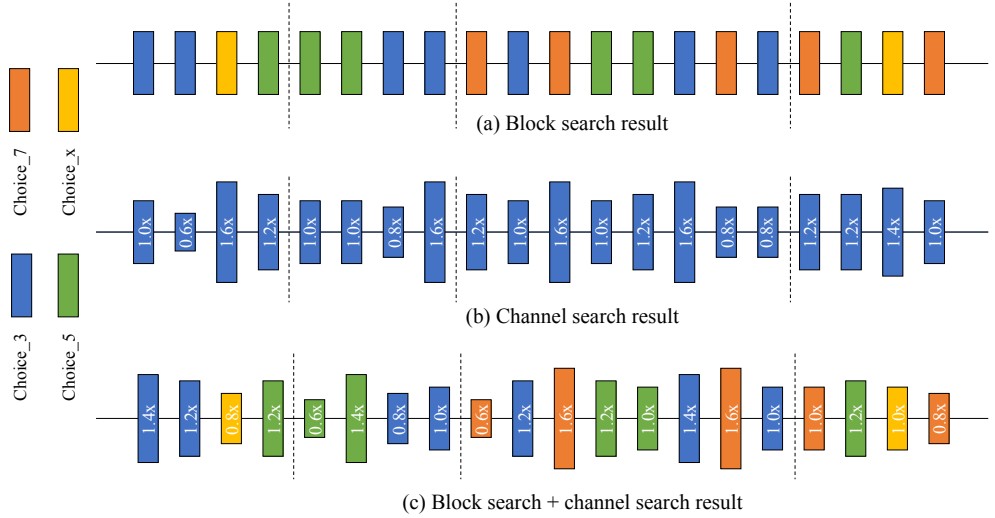

Figure 7: Structures of searched architectures in Sec. 4. (a) Result of building block search. (b) Result of channel search on all_choice_3 structure. (c) Result of channel search on best building block search structure.

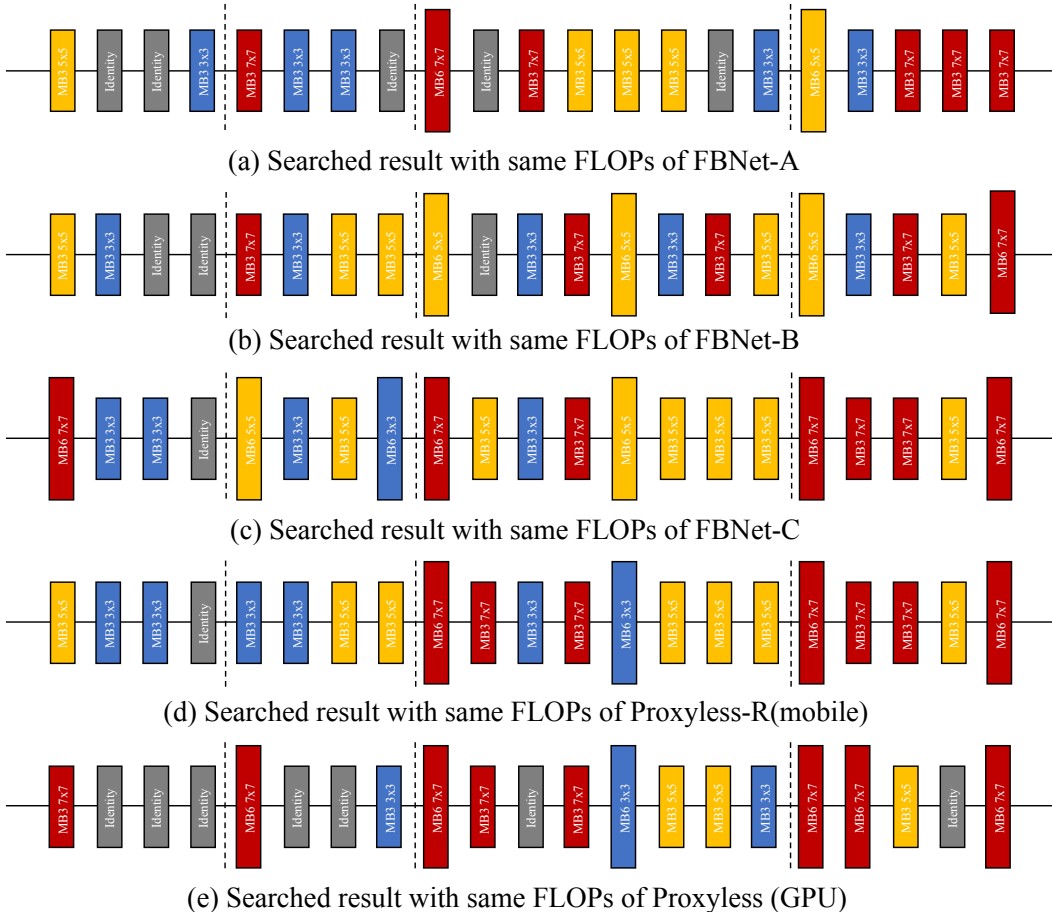

(a) Searched result with same FLOPs of FBNet-A

(b) Searched result with same FLOPs of FBNet-B

(c) Searched result with same FLOPs of FBNet-C

(d) Searched result with same FLOPs of Proxyless-R(mobile)

(e) Searched result with same FLOPs of Proxyless (GPU)

Figure 8: Structures of searched architectures under FLOPs constraints by using *ProxylessNAS* search space, see Table 4 for details.

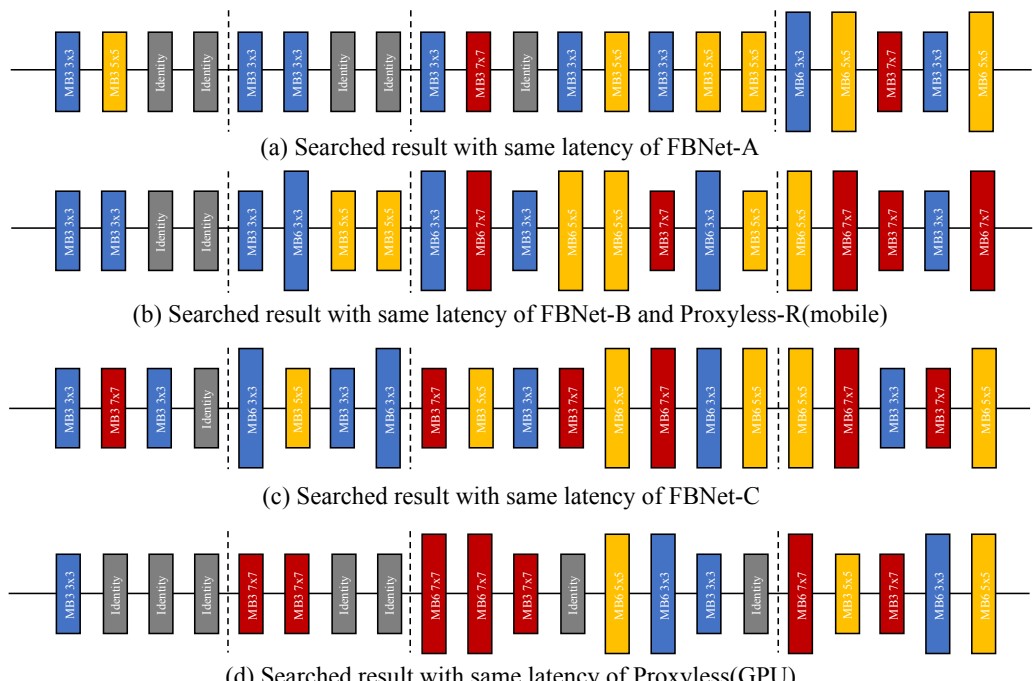

(a) Searched result with same latency of FBNet-A

(b) Searched result with same latency of FBNet-B and Proxyless-R(mobile)

(c) Searched result with same latency of FBNet-C

(d) Searched result with same latency of Proxyless(GPU)

Figure 9: Structures of searched architectures under GPU latency constraints by using *ProxylessNAS* search space, see Table 4 for details.

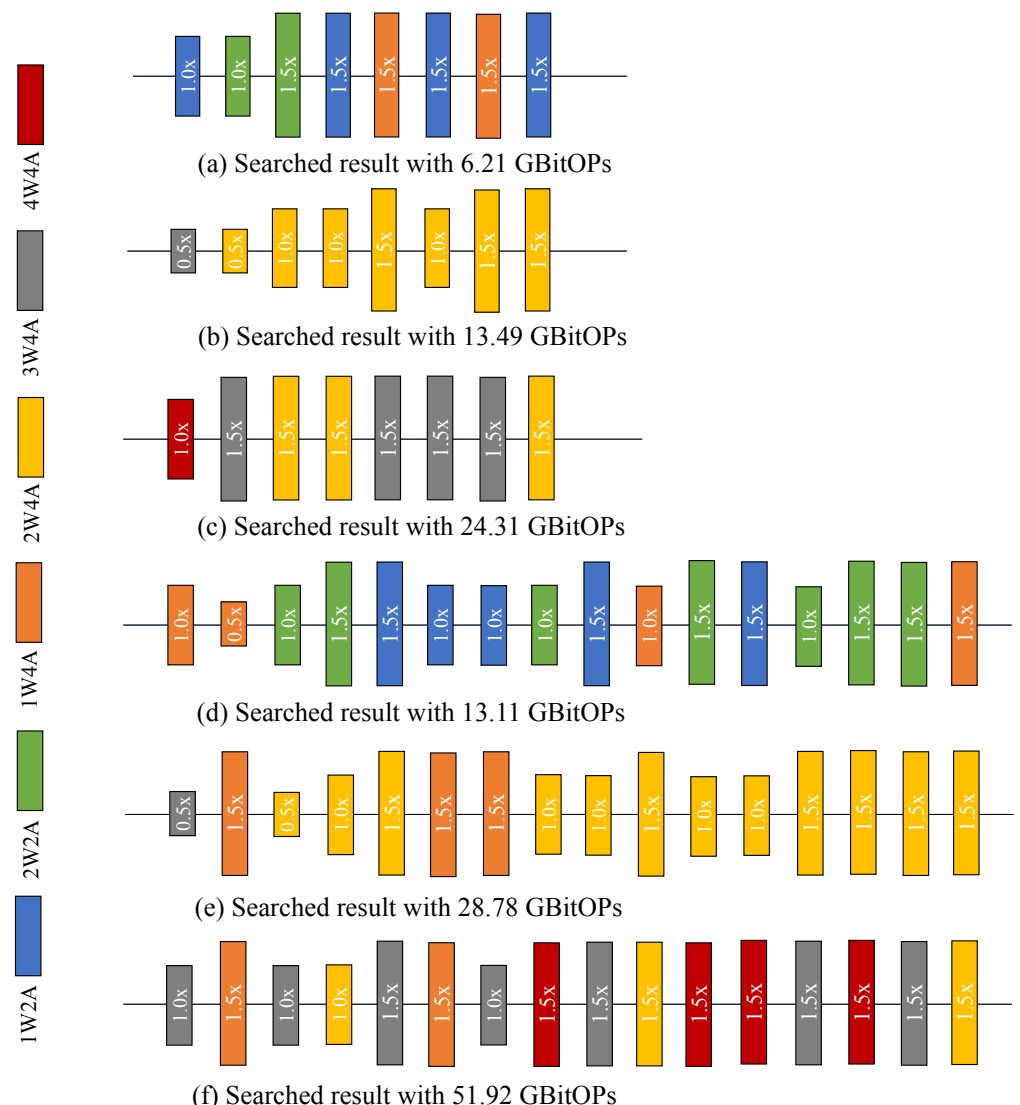

(a) Searched result with 6.21 GBitOPs

(b) Searched result with 13.49 GBitOPs

(c) Searched result with 24.31 GBitOPs

(d) Searched result with 13.11 GBitOPs

(e) Searched result with 28.78 GBitOPs

(f) Searched result with 51.92 GBitOPs

Figure 10: Searched architectures of joint searching channel size and bit width under BitOPs constraints, see Table 5 for details. (a) - (c) are searched based on Resnet18. (d) - (f) are searched based on Resnet34.

