# OpenReview forum: "SINGLE PATH ONE-SHOT NEURAL ARCHITECTURE SEARCH WITH UNIFORM SAMPLING"
_ICLR.cc/2020/Conference — Reject_

### Official Review · AnonReviewer2 · 2019-10-21
**Official Blind Review #2**

**Rating:** 8

**Review:**

This paper presents a new one-shot NAS approach. Parameter updating and structure updating are optimized separately. In the process of parameter optimization, different from the previous methods, the network samples the structure according to the uniform probability distribution. This sampling method can avoid the coupling caused by optimizing the structural parameters at the same time. After training the supernet, the network uses the genetic algorithm to search the structure.

Pros:
+ This paper is well-written and easy to follow.
+ The method proposed in this paper is very efficient in the search stage and saves memory. During the searching stage, the constraints of the model can be restricted by the genetic algorithm.
+ This method could directly search architectures on the large scale dataset.
+ The results are promising. Experiments are performed on various structures and including quantization layers.

Still, I have several minor concerns regarding the algorithm and experiments.

1. Why does the author think that the supernet needs to be trained by single path all the time? In the architecture searching space, under the same computational constraints, some models perform well, and some do not. Why do the good models and the worse models use the same probability to sample? Wouldn't it require much more time to optimize the supernet? In the experiment part, the author compares a related work FBNet, which uses the structure parameter \theta and temperature parameter \tau to control the sampling. When the temperature \tau is high, the distribution of the samples degenerates into the single path sampler, and the accuracy is used to optimize the parameter \theta, which makes \theta tend to select the well-performed models. With the decrease of the temperature \tau, the probability of sampling well-performed architectures is higher than the worse-performed models. Therefore, what is the advantage of the single path sampling method over \tau controlled sampler? It seems that gradually pruning the worse-performed search space would accelerate searching time. More analysis is preferred rather than the superior results from the experiments.

2. Is identity in the search space? For at the last of Page4, the author said 'our choice block does not have an identity branch', and the listed architecture on page 13 has the identity.

3. Is the supernet training for 120 epochs necessary? Is the rank of network structures become steady when training for 120 epochs. Because for the subsequent EA-based network structure search, the loss scale is not important, but the sorting (ranking) is important. (This beyond the scope of this paper, but relates to the NAS area)

4. What is the number of parameters for all the listed models in Tab. 3 & Tab. 4? Listing the parameters makes it easier for the followers.

5. What kind of structure do the mixed-precision networks use? The original resnet module, or the bireal resnet module [r1]?

6. While recalculating the statistics of all the BN,  is backpropagation required or just run the inference without gradient backpropagation.

7. The author uses the approximate complete set of Imagenet to train supernet. If we directly inherit the parameter from the supernet, can we accelerate the training of searched good structure? Will Top-1 acc lower than training from scratch?

[r1] Bi-real net: Enhancing the performance of 1-bit cnns with improved representational capability and advanced training algorithm


**Experience Assessment:**

I have published in this field for several years.

**Review Assessment: Checking Correctness Of Derivations And Theory:**

I carefully checked the derivations and theory.

**Review Assessment: Checking Correctness Of Experiments:**

I carefully checked the experiments.

**Review Assessment: Thoroughness In Paper Reading:**

I read the paper thoroughly.

---

> ### Author Response · Authors · 2019-11-07
> **Response**
>
> Thanks for your comments. And there are the answers below to your every concerned question.
>         1. On the one hand, the model converges quickly may not achieves better performance at final. It is hard to distinguish the good models from worse models correctly at the beginning of training. On the other hand, Single path with uniform sampling is parameter-free, which is also an advantage for practical use. I believe that adjusting the temperature carefully in FBNet[1] can also achieve good results under specific constraint. But it is inconveniently to search multi architectures with different constraints because of needing to adjust the hyper parameters carefully.
>         2. The identity exists in the search space of ProxylessNAS[2], not in our shufflenet-like choice blocks. The architectures in page 13 is the found results based on search space of ProxylessNAS[2].
>         3. Yes. Training the superset with 120 epoch may not be necessary. But it is hard to decide how long should the supernet to be trained because we don’t know when the rank is steady.
>         4. We have listed the parameters of models in Tab.3 and Tab.4 in the revised version.
>         5. We used the ReLU-only pre-activation Resnet module from [3]. This module was also used in DNAS[4].
>         6. Just run the inference without gradient back propagation.
>         7. We choose to train from scratch for fair comparison. Maybe the fine-tuning can accelerate the training of searched architecture. But the performance of architecture will be influenced by the training setting of fine-tuning. The accuracy of fine-tuning is higher or lower, we think it depends on the training setting of fine-tuning.
>
>
> [1] Bichen Wu, Xiaoliang Dai, Peizhao Zhang, Yanghan Wang, Fei Sun, Yiming Wu, Yuandong Tian, Peter Vajda, Yangqing Jia, and Kurt Keutzer. Fbnet: Hardware-aware efficient convnet design via differentiable neural architecture search. arXiv preprint arXiv:1812.03443, 2018a.
> [2] Han Cai, Ligeng Zhu, and Song Han. Proxylessnas: Direct neural architecture search on target task and hardware. arXiv preprint arXiv:1812.00332, 2018.
> [3] Kaiming He, Xiangyu Zhang, Shaoqing Ren, and Jian Sun. Identity mappings in deep residual networks. In European conference on computer vision, pp. 630-645. Springer, 2016b.
> [4] Bichen Wu, Yanghan Wang, Peizhao Zhang, Yuandong Tian, Peter Vajda, and Kurt Keutzer. Mixed precision quantization of convnets via differentiable neural architecture search. arXiv preprint arXiv:1812.00090, 2018b.

---

### Official Review · AnonReviewer1 · 2019-10-25
**Official Blind Review #1**

**Rating:** 6

**Review:**

Authors revise the one-shot NAS algorithm in this work. One-shot NAS that employs a supernet to share the weights between subnets is an efficient NAS algorithm. Authors develop a new training paradigm to train the supernet sufficiently. Specifically, they uniformly sample a single path from supernet at each iteration to make the training effective and stable.

Pros:
1.	Improve the one-shot NAS by uniform path sampling.
2.	Experiments demonstrate effectiveness.

Cons:
1.	Authors only report the performance of the best architecture after fine tuning. It is interesting to see the performance of subnets after obtaining the supernet. Is the better subnet in supernet still better than others after fine tuning?
2.	Given the large number of single paths, it is hard to train each one sufficiently within a supernet. Authors may demonstrate how one-shot NAS can address the problem.


**Experience Assessment:**

I have read many papers in this area.

**Review Assessment: Checking Correctness Of Derivations And Theory:**

I assessed the sensibility of the derivations and theory.

**Review Assessment: Checking Correctness Of Experiments:**

I assessed the sensibility of the experiments.

**Review Assessment: Thoroughness In Paper Reading:**

I read the paper at least twice and used my best judgement in assessing the paper.

---

> ### Author Response · Authors · 2019-11-07
> **Response**
>
> Thanks for your comments.
>         1. In our experiments, the rank performance of superset is good enough to find the superior architectures. It may not pick out the best model among many similar models correctly, but can distinguish the superior models. But I think it still depends on search space. While the search space becomes more complicated, the rank performance will decrease. This is also what we aim to improve in the future.
>         2. Assume there are n options and L layers in superset. If we train the supernet with T iterations, then each option is expected to be trained with T/n iterations due to the uniform sampling. And it is irrelevant to L due to the weight sharing in supernet. Generally, n will not be a huge number in practice. We trained a superset with n=32 options (combining the block and channel search space) and it can still converge well. Of course, if the options increase significantly, it’d better to train the superset with more iterations. We know single path training can not cover all paths in the superset, but can train each options sufficiently.

---

### Official Review · AnonReviewer3 · 2019-10-31
**Official Blind Review #3**

**Rating:** 6

**Review:**

*UPDATE* I have read the other reviews and authors' responses. All the reviewers agree that improving single-shot NAS is an important problem, and that sampling single-paths can be a plausible approach for it that avoids weight coupling. Consequently, I have updated my rating to weak accept. I think the paper can be substantially stronger though.
The key claim that set this paper apart from other single-path NAS approaches is that they use a fixed distribution (in particular, the uniform distribution) to sample from unlike others like FBnet who use a trainable distribution. They argue that uniform is parameter-free whereas trainable distributions introduce additional parameters that need to be trained. Their findings motivate a natural follow-up question: could we use a different fixed distribution? Reviewer2 has a similar question in their review. Perhaps a distribution that is weighted according to (some proxy of) how much computational resource the networks take to train? Could such a distribution also be parameter-free and give good benefit over uniform distribution without needing to be updated during supernet training? Such an analysis of the prior distribution will make the paper even stronger.


The paper studies a sequential optimization approach to neural architecture search that can provide some benefit over nested or joint approaches. The core challenge in sequential approaches (which first train the weights of a supernetwork; then search through possible architectures which inherit appropriate weights from the supernetwork) is that the weights for a giant network may not be optimal for the weights of a sub-network encountered during subsequent architecture search. The core benefit of such an approach compared to nested approaches is that the subsequent search phase only needs to perform network inference with inherited weights; not train a sub-network from scratch.
The primary contribution is to fix a prior distribution over architectures and sample from them when training the supernetwork. This simple fix helps the weights of the giant network be more useful when inherited into any sub-networks during architecture search.
The paper will be substantially stronger with a careful study of the choice of the prior distribution and how it affects (a) the rejection sampling step needed to ensure the sampled architectures satisfy complexity constraints, and (b) the performance of the eventual neural architecture search procedure.
Another experiment that will be valuable is to rigorously validate the hypothesis that reducing weight coupling in the supernetwork training is crucially linked to improving the downstream architecture search performance.

Minor (writing comments):
Introduction: "Complex optimization techniques are adopted." This statement is awkward. Does this paper specifically adopt more complex methods to address the shortcomings of gradient methods. Or, does the community broadly research more complex methods as a consequence (and you are advocating for a return to simpler gradient methods)?
Pg5: "we randomly choice a range" -> "choose"


**Experience Assessment:**

I do not know much about this area.

**Review Assessment: Checking Correctness Of Derivations And Theory:**

I assessed the sensibility of the derivations and theory.

**Review Assessment: Checking Correctness Of Experiments:**

I assessed the sensibility of the experiments.

**Review Assessment: Thoroughness In Paper Reading:**

I read the paper at least twice and used my best judgement in assessing the paper.

---

> ### Author Response · Authors · 2019-11-07
> **Response**
>
> Thanks for your comments.
>         1. I read the comments thoroughly and find that you give a detailed summary of our paper. It seems that you have understood the idea and the contribution of our paper from your comments. But I didn’t see any problem about the paper was pointed out except the minor writing problems.  If you have any concerned question, it is our pleasure to give you an answer.
>         2. As for the writing problems, we have revised them. Thanks for pointing out that.

---

> > ### Comment · AnonReviewer3 · 2019-11-07
> > **Pointing out 2 specific weaknesses**
> >
> > Please include analysis of the choice of the prior distribution. How does it affect (a) the rejection sampling step needed to ensure the sampled architectures satisfy complexity constraints, and (b) the performance of the eventual neural architecture search procedure?
> >
> > Please design a rigorous experiment to validate the hypothesis that reducing weight coupling in the supernetwork training is crucially linked to improving the downstream architecture search performance.

---

> > > ### Author Response · Authors · 2019-11-08
> > > **Response**
> > >
> > > Thanks for your comments.
> > >         1. As we explained in the paper(See Section 3), we empirically find that uniform sampling is good. When using uniform sampling, we didn’t require the sampled architectures to satisfy the complexity constraint. We also experimented with a variant that samples the architectures uniformly according to their constraints, named uniform constraint sampling. We force the sampled architectures to be uniform among the ranges, which performs slightly better than uniform sampling on searching multiple architectures satisfying different constraints. Maybe we didn’t express well in the paper, we have revised it.
> > >         2. Please see the Fig.1 which compared the method in [1] with our single path strategy. The method in [1] proposed to sum up all operations within a layer, which deeply coupled weights in supernet. So they proposed a drop path strategy to alleviate the problem. But it is not good enough. In our paper, we propose our single path strategy to decouple the weights extremely. So the results in Fig.1 can not only show that drop rate matters a lot, but also verify the benefit of weight decoupling by our single path strategy. It is our duty that we didn’t explain these clearly in the paper. We have revised them in the paper. Thank you for pointing out that.
> > >
> > > [1] Gabriel Bender, Pieter-Jan Kindermans, Barret Zoph, Vijay Vasudevan, and Quoc Le. Understanding and simplifying one-shot architecture search. In International Conference on Machine Learning, pp. 549–558, 2018.

---

### Public Comment · ~Zhixing_Chen1 · 2019-10-16
**Discussing with previous "single path" NAS works**

Thanks for this interesting work. Since the title is single path one-shot NAS, I assume the "single path" is one of the main contributions of this paper.

Does "Single path" indicate when optimizing the shared paremters, only one candidate network is used to forward and only its parameters are optimized? I found that some previous NAS methods also proposed and used the similar idea of "single path".
For example, [1] and [2] use discrete architectural parameters during forwarding, which equals to "single path". Other similar works include [3] and [4].
A special case of [5] is also "single path" (See 3.2., when they dropping k-1 inputs for each node).

Do the authors mind discussing with the abovementioned "single path" NAS works.

[1] Differentiable Neural Architecture Search via Proximal Iterations (arXiv1905)
[2] Searching for A Robust Neural Architecture in Four GPU Hours (CVPR 2019)
[3] Single-Path NAS: Designing Hardware-Efficient ConvNets in less than 4 Hours (arXiv 1904)
[4] Universally Slimmable Networks and Improved Training Techniques (arXiv1903, ICCV19)
[5] Understanding and Simplifying One-Shot Architecture Search (ICML 2018)

---

> ### Author Response · Authors · 2019-10-16
> **Discussing with previous "single path" NAS works**
>
> Thanks for your comments.
>
> As we explained in our paper(see Section 3).
> "We note that sampling a path according to architecture distribution during optimization is already used in previous weight sharing approaches. The difference is that, the distribution Γ(A) is a fixed priori during our training (Eq. (7)), while it is learnable and updated (Eq. (3)) in previous approaches". So do the papers [1][2][3] you mentioned. Besides, The "single path" in [3] means their single path search space (superkernel), not the single path sampling.
>
> [4] proposed to sample number of width scales to scale the channel of the full model, and trained the model by accumulating the gradients of all paths. It may be a kind of single path, but it is restricted because all the layers were applied to the same width scale, and it is not uniform sampling.
>
> As for [5], they proposed a path dropout strategy for each layer to train the supernet. Experiments in [5] indicated that the training is very sensitive to the dropout rate. Maybe there are some views that our single path is the extreme of path dropout strategy, but it's not trivial as it is based on the non-trivial observation that reducing the weight coupling in the supernet is beneficial. And we did bring benefits for supernet training. Besides, it's hyperparameter-free. We have explained it in our paper, please refer to the Section 3 for details.
>
> In the end, thanks again for your attention. Maybe there are some concurrent works that are similar to our single path and we didn't mention all of them in our paper. We will update our paper in the future if need be.

---

> > ### Public Comment · ~Dimitrios_Stamoulis1 · 2019-11-02
> > **Discussing with previous "single path" NAS works**
> >
> > Thanks for this interesting work. It is really exciting to see single-path-based formulations being enhanced with insightful sampling methods.
> >
> > To reiterate on this: the part you quoted from the paper focuses on comparing against multi-path approaches only! However, as Zhixing correctly pointed out, since the earlier ProxylessNAS-like methods, several single-path formulations have been explored, hence necessitating a discussion from the authors on how their method improves/compares against existing formulations.
> >
> > In fact, early Single-Path approaches have been previously published (ECML'19 paper: https://ecmlpkdd2019.org/downloads/paper/880.pdf ) and currently achieve results higher than the accuracies reported in this submission, while targeting on-par search space (e.g., single path with top1 75.2% in https://arxiv.org/pdf/1907.00959.pdf ) compared to the 74.7% reported in the paper.
> >
> > Such discussion and citations seem reasonable to appear in the 'Related work' and/or the 'Comparison with State-of-the-arts.' sections.

---

> > > ### Author Response · Authors · 2019-11-07
> > > **Discussing with previous "single path" NAS works**
> > >
> > > Thanks for your comments.
> > >         As we explained in the paper(See Section 3), the difference is that, the distribution of our single path is a fixed priori (uniform sampling) during our training, while it is learnable and updated in previous approaches.  Our single path with uniform sampling can ensure that different architectures in our supernet are updated at the same level, which makes the subnets being comparable, and avoid the “dilemma of exploitation and exploration” problem during training. These discussion can be found in our paper(See Section 2).
> > >         As for the papers you mentioned, their single path strategy still belongs to the method whose distribution is learnable and updated during training, and is different from ours. However, since they are concurrent works, we didn't know and discuss them in our paper, we will add them in our future version.
> > >         We also retrain the found architecture (74.96) in https://ecmlpkdd2019.org/downloads/paper/880.pdf with the same training setting in our paper, and we only achieve 74.6. And we find that the found architecture in https://arxiv.org/pdf/1907.00959.pdf already involve SE( Squeeze-and-Excitation) block, so it is non-comparable to ours.
> > >
> > > In the end, thanks again for your attention.

---

### Decision · Program_Chairs · 2019-12-19

**Decision:**

Reject

**Comment:**

This paper introduces a simple NAS method based on sampling single paths of the one-shot model based on a uniform distribution. Next to the private discussion with reviewers, I read the paper in detail.

During the discussion, first, the reviewer who gave a weak reject upgraded his/her score to a weak accept since all reviewers appreciated the importance of neural architecture search and that the authors' approach is plausible.
Then, however, it surfaced that the main claim of novelty in the paper, namely the uniform sampling of paths with weight-sharing, is not novel: Li & Talwalkar already introduced a uniform random sampling of paths with weight-sharing in the one-shot model in their paper "Random Search and Reproducibility in NAS" (https://arxiv.org/abs/1902.07638), which was on arXiv since February 2019 and has been published at UAI 2019. This was their method "RandomNAS with weight sharing".

The authors actually cite that paper but do not mention RandomNAS with weight sharing. This may be because their paper also has been on arXiv since March 2019 (6 weeks after the one above), and was therefore likely parallel work. Nevertheless, now, 9 months later, the situation has changed, and the authors should at least point out in their paper that they were not the first to introduce RandomNAS with weight sharing during the search, but that they rather study the benefits of that previously-introduced method.

The only real novelty in terms of NAS methods that the authors provide is to use a genetic algorithm to select the architecture with the best one-shot model performance, rather than random search. This is a relatively minor contribution, discussed literally in a single paragraph in the paper (with missing details about the crossover operator used; please fill these in). Also, this step is very cheap, so one could potentially just run random search longer. Finally, the comparison presented may be unfair: evolution uses a population size of 50, and Figure 2 plots iterations. It is unclear whether each iteration for random search also evaluated 50 samples; if not, then evolution got 50x more samples than random search. The authors should fix this in a new version of the paper.

The paper also appears to make some wrong claims in Section 2. For example, the authors write that gradient-based NAS methods like DARTS inherit the one-shot weights and fine-tune the discretized architectures, but all methods I know of actually retrain from scratch rather than fine-tuning. Also, equation (3) is not what DARTS does; that does a bi-level optimization.
In Section 3, the authors say that their single-path strategy corresponds to a dropout rate of 1. I do not think that this is correct, since a dropout rate of 1 drops every connection (and does not leave one remaining). All of these issues should be rectified.

The paper reports good results on ImageNet. Unfortunately, these may well be due to using a better training pipeline than other works, rather than due to a better NAS method (no code is available, so there is no way to verify this). On the other hand, the application to mixed-precision quantization is novel and interesting.

AnonReviewer2 asked about the correlation of the one-shot performance and the final evaluation performance, and this question was not answered properly by the authors. This question is relevant, because this correlation has been shown to be very low in several works (e.g., Sciuto et al: "Evaluating the search phase of Neural Architecture Search" (https://arxiv.org/abs/1902.08142), on arXiv since February 2019 and a parallel ICLR submission). In those cases, the proposed approach would definitely not work.

The high scores the reviewers gave were based on the understanding that uniform sampling in the one-shot model was a novel contribution of this paper. Adjusting for that, the real score is much lower and right at the acceptance threshold. After a discussion with the PCs, due to limited capacity, the recommendation is to reject the current version. I encourage the authors to address the issues identified by the reviewers and in this meta-review and to submit to a future venue.

---

> ### Author Response · Authors · 2019-12-27
> **Response**
>
>         Although the final decision is made, we still want to clarify some controversial opinions from the meta-review comments.
> 	Our work and RandomNAS are parallel work. And we did research and discover the method independently. Our work is inspired by the “Understanding and simplifying one-shot architecture search” (http://proceedings.mlr.press/v80/bender18a/bender18a.pdf). But the existing one-shot approach suffers the weight coupling problem and is sensitive to the dropout rate during supernet training, so we propose a single path supernet with uniform sampling to alleviate these problems. The meta-review said that the only real novelty of our work is to use a genetic algorithm rather than random search, we don’t think it is true.
>         First, we propose a single path supernet and uniform sampling to alleviate the weight coupling problem and avoid adjusting the hyper-parameter dropout rate during supernet training. We also carry out experiments to verify this proposal, please see Figure.1. By the way, the meta-review said that our single path strategy doesn’t correspond to a dropout rate of 1 since a dropout rate of 1 drops every connection. Actually, we have explained this situation clearly in our paper (“Note that the original drop path in (Bender et al., 2018) may drop all operations in a block, resulting in a short cut of identity connection. In our implementation, it is forced that one random path is kept in this case since our choice block does not have an identity”). As Figure.1 shows, we can find that the dropout rate is really very sensitive to the supernet training and our method performs better. Second, we propose that random search is not effective to select the architecture with best performance for a large search space, and this is why we propose to use an evolutionary algorithm. In order to verify this, we also carry out experiments to compare these two algorithms fairly. the meta-review thought “this step is very cheap, so one could potentially just run random search longer”, but we don’t think so. The efficiency and effectiveness of search algorithm are different. In theory, searching all architectures in search space can achieve the best result, but the efficiency is too low for us to do that, so we need heuristic algorithms to achieve the local optimal result. If we run random search longer to get a good result, we can always run evolutionary search as longer as random search to get a better result. The meta-review also had a suspicion that our comparison is unfair. It is our duty that missing some detailed description in our paper. But the comparison is totally fair. The number of random selected architectures in each iteration is 50 and the total number of iterations is 20, which is the same as evolutionary algorithm. Figure. 2 plots the top-10 of both algorithm and we can find see the evolutionary algorithm performs better. Third, the design of our choice block is also novel. We propose a new choice block based on weight sharing for channel search. And as the meta-review said, the application to mixed-precision quantization is novel, too.
> 	On the other hand, the meta-review questioned that our good result on ImageNet may be due to using a better training pipeline than other works. We think it is unreasonable. We have emphasized that we use the same settings (including data augmentation strategy, learning rate schedule, etc.) as (Ma et al., 2018) to retrain all architectures. What’s more, in order to eliminate the difference of search spaces, we also benchmark our approach to the same search space. We have explained these clearly in our paper. (“To make direct comparisons, we benchmark our approach to the same search space of (Cai et al., 2018; Wu et al., 2018a). In addition, we retrain the searched models reported in (Cai et al., 2018; Wu et al., 2018a) under the same settings to guarantee the fair comparison.”). And we have released our code on : https://github.com/megvii-model/SinglePathOneShot.
> 	As for the wrong description in Section 2, we will revise them.
> 	For last question about the correlation of the one-shot performance, the meta-review quoted "Evaluating the search phase of Neural Architecture Search" (https://arxiv.org/abs/1902.08142) saying that the correlation has been shown to be very low in several works. We believe that it depends on search space. While the search space becomes more complicated, the rank performance will decrease. In our experiments, we find that our method is good enough to find the superior architecture. That is why we can achieve better result on ImageNet under the fair comparison with other methods.